# Semantic web, social data as a resource and optimal data governance structures*

Juan G. Diaz Ochoa[1,*,†], Elena Ramirez[2,†]

[1] *PerMediQ GmbH, Germany*

[2] *Bürger Stiftung, Germany*

## Abstract

Recently the development of technologies in semantic web have contributed to decentralized data generation and storage, dissolve data silos and provide customers data sovereignty. This has opened the perspective to bring forward the concept of social data, i.e. data that is used with a social purpose. Despite social data is now an established science to analyze human behavior, there is a lack of precise definitions about social data as a resource, and its implications when it is industrially exploited. The goal is to exactly define how data can be considered as a resource (including the infrastructures required for social data), and what is the role of this resource in the society. We analyze the problems associated to current data-infrastructures and discuss possible solutions. We conclude that social data not only require a special infrastructure but also appropriate forms of governance. We provide some potential use-cases where infrastructures of social data can be applied.

**Keywords**

Social data, Data governance, Data cooperatives

## 1. Introduction and formal definitions

Data, economics and society are closely related [1]. However, despite its social relevance, seldom the social character of data and its implication in social structures (including mathematical aspects) are tacked into account.

Social data is usually considered scientifically, since it promises to learn about "what the world thinks" about a specific issue, brand, celebrity, or other entity. In addition, it enables better decision-making in a variety of fields including public policy, healthcare, and economics [2], [3]. Although social data is an extremely significant scientific field, the manner in which it is collected and stored is equally relevant. Our objective is to examine social data as a resource and its relation to data infrastructures and business models. Observe that social data, both as data and as a resource, can be used to gain insight into human behavior and human interaction[2].

*NeXt-generation Data Governance 2024 workshop*

Corresponding author.

† These authors contributed equally.

✉ juan.diaz@permediq.de (J.G. Diaz Ochoa); elena.ramirez@harryslab.de (E. Ramírez Barrios)

🆔 0000-0002-9893-4068 (J.G. Diaz Ochoa); 0000-0002-4250-2047 (E. Ramírez Barrios)

Although data is regarded as a raw resource that can be extracted and processed as any other resource (including precious commodities like gold and oil), it is also fundamental to defining our own identity, our social cohesion and to our understanding of the environment as a whole. When citizens/customers are seen as both resources and people, the concept of data as a resource and the concept of data shaping our lives are contradictory. Semantic-web seem to be a way to solve this contradiction, since i.) it facilitates the use of meta-data (data about data) as semantic annotations, ii.) makes use ontologies to describe knowledge needed to understand collections of web information, and iii.) makes use logic-based techniques to process and query collections of meta-data and ontologies [4].

In particular developments in semantic web, like *Solid* [5] or the *Gaia-X* [6], enable social data as technology. Thus, according to this technological paradigm, social data is not simply contained in a database. Instead, it has a structure that facilitates interlinking with different data sources from different customers. From a technical perspective, this allows the straightforward definition of graph and knowledge-databases. Furthermore, such data is not stored in silos, and has mechanisms that enable decentralized storage schemas, with decentralized and customer-oriented governance. Moreover, customers are able to authorize or disapprove the use of their information by companies in a variety of ways. Finally, this kind of data recording facilitates the study of human behavior and human interaction.

However, there is a lack of fundamental axioms aiming at providing a clear conceptual basis for both use and value creation of social data. For the definition of these principles there are three aspects to be considered: the philosophical basis of internet, the mathematical basis, and the socioeconomic aspect of data.

## 2. Fundamental axioms defining data as resource

The definition about what is data is relevant to understand its value and thus its impact in economy. The fundamental axiom is that:

> *Data is essentially useless, and that its value only arises when it is used to create information, for instance in form of statistical models.*

Data is not per se information [7]. There are several methods to process data in order to generate information, like creating a writing description and interpretation from the input data or by using gathered data to construct mathematical representations and models. The obtained information can be, for example, the analysis of individual consumer behavior.

Current paradigms consider data as something that should be extracted to obtain information. Thus, data obtention has become an extractive industry aiming at exploiting data as a raw resource. However, data can both stored (for retrospective analysis) or be continuously generated (for real world analysis). Thus, data is essentially not like gold (which is an element

and is thus eternal) or oil (which is created from organic matter in very slow processes). Data can always be generated and re-generated. Data is not static, and can be produced considering initial hypotheses, which implies that data is not objective. Thus, data production depends on starting hypotheses and experimental adjustments. Finally, an important feature, that differentiates Data from Oil is that Data are not exhausted. Data can be used many times and also by many users at the same time. Their use is not exclusive but restricted.

Any form of data production requires an initial hypothesis (experimental construction). For instance, the recording of wheat deposits in ancient Sumerian or Egyptian records in cuneiform tables had an administrative character. Such data is however used today in archaeology and to better understand the status of societies in the bronze age [8]. Thus, data has been relevant in any society but was not harmonically and systematically recorded. Data can have any form, one of them is numerical. In general, any kind of data is unstructured, while numerical data is usually provided in a structural way. Finally, and most important, data can always be new defined and regenerated, not only externally but also by everyone.

Therefore, data is not simply a raw material like oil (which has been repeatedly used as a metaphor, [9]), but is the fundamental resource required to create information that help us to relate with our neighbors and environment. Data can be stored, as well as deleted, associated to other data, and recombined. Data can be generated de novo, but data can also be generated from other data. Thus:

*Data is fluid and is not invariant and is essentially an asset* [10].

Only after the development of computational methods and internet was possible to generate large amounts of data across different societies. From the first axiom this implies that such infrastructure leads to the generation of information in global scale used in decision making, either politically or in business. This fast development has shown that information in small scale is limited.

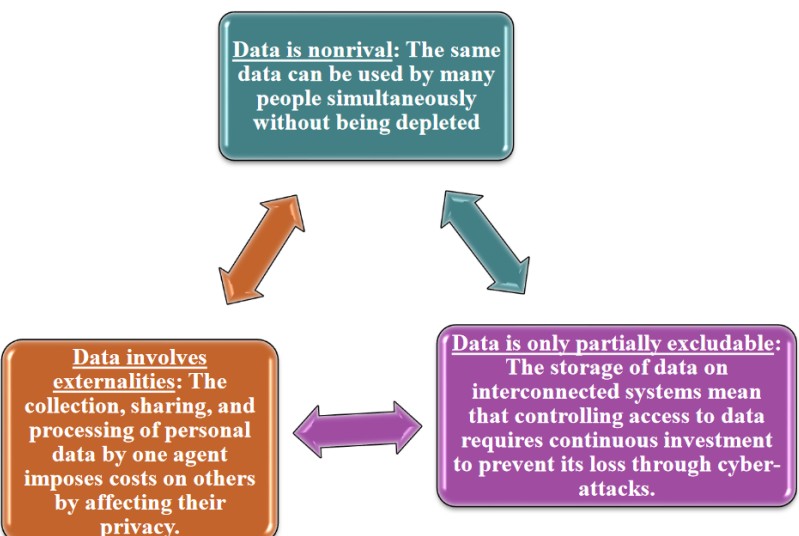

**Figure 1:** Data cannot be trivially defined, and its definition cannot be reduced in an objective way. Instead, data depends on externalities, cooperative use (non-rivalry) and execution [figure extracted from the slides "The economics of data", presented at the SoSy2023 [11]].

Additionally, data (even unstructured data) is often stored or transformed into numerical values. This implies that any representation based on these data will be constrained in terms of its qualities and structure into a numerical description. In addition, any information derived from data will be based on mathematical - numerical models. Nowadays, information is designed and delivered by providers, either from universities and large research institutes, or from enterprises. Information providers profit from the generation and processing of information, offering new business opportunities for companies and enterprises. In a nutshell, data is much more complex than a reductionist metaphor trying to associate data with a resource. Its dependence on externalities, execution and non-triviality should always be considered (see Figure 1).

After introducing a definition of data, it is now relevant to consider the problem of its governance and societal impact through this governance.

## 3. Data governance and societal problems

Despite individuals gathering and analyze individual information, data collection and information generation are processed for large organizations (government - corporations) or for very small scales – individual users, for example in search machines. Information generation is based on the statistical analysis of large amounts of data to create reliable statistical and mathematical models at different scales. Due to the size and scale of the processed data, extensive computational infrastructures are required.

This explains why big companies dominate the market. Such companies are currently dealing with social data, considering that all the gathered data contains traces of human behavior. However, paradoxically the current infrastructure implemented to deal with this data is not social. The implemented architecture considers first the collection of data, and then the possibility to relate this data to behavioral patterns or to relate this data with other data sources.

Furthermore, it is not only the challenge in constraining social and real data to pre-defined data structures, but also inherent paradox of storing inherent social data (from individuals embedded in an environment and a society) in silos. Seldom information is processed for small communities. This has created problems of data governance, where individual data is exploited as raw material to deliver services. Enterprises seek to bring individuals to behave themselves as raw material to produce even more data, bringing typical problems in data extraction like privacy [12].

Increasing revenue or creating strategic societal advantage is the goal of governments and enterprises, as well as individuals. As a result, data as a raw material is monopolized. Despite data is not a raw material, its current use and exploitation have finally turned it into a raw material. Thus, current forms of data governance have generated two main problems.

## 3.1. Problem 1: Data governance and the structure of social networks vs. social structures

Technological infrastructure tends to have a scale free distribution. This one is due the fact that centralized structures are more convenient to maintain infrastructures. This explains why only few providers dominate all the information traffic, making such companies very valuable [13].

Accordingly, data generation and processing obey the design of infrastructures, including social data, which is currently gathered by platforms designed to manage social networks.

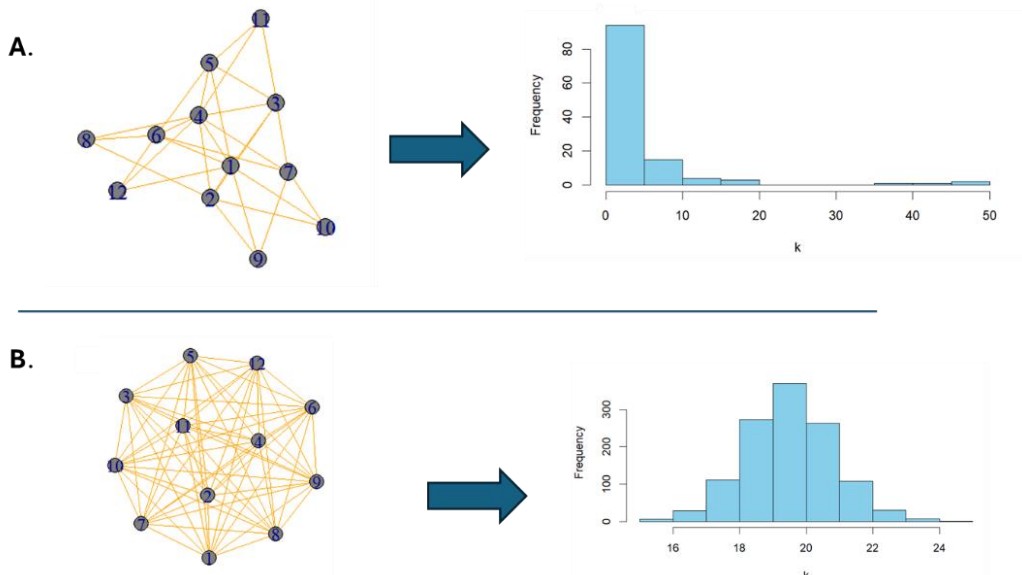

**Figure 2:** Technical and data-processing networks (for instance social networks) are usually based in a scale-free distribution constraining social behavior to such structure (A). However, societies tend to organize with a log-norm distribution (B).

Such infrastructures constrain the data traffic, as well as the way how social interactions are governed and guided (using for instance algorithms to control and optimize the attention of customers) in order to make such companies profitable. Thus, the underlying structure generated by data infrastructures tends to be scale-free. Such constraints as well as the apparent uniqueness of scale free distributions has let to consider this characteristic as a kind of elegant law of complex systems [14]. However, real societies do not show a scale free distribution, but rather a log-norm distribution [15] (see Figure 2).

In the way how information is processed and governed influences the way how individuals and societies relate. Furthermore, scale free structures are helpful to create monopolies – under the assumption that scale freeness is more robust against failure. This however possess a problem, since private companies are currently controlling a fundamental infrastructure that deals with the communication of individuals in a society.

Additionally, companies create incentives to maintain social networks constrained to scale-free structures. Incentives are recognition (economy of attention) and the possibility to have a global coverage to promote ideas and products. Is a kind of sense of universality. Thus, social data exploited by enterprises create economic and behavioral incentives that generates a particular topology but distort natural ways of social connectivity and communication between individuals (Balick, 2023).

### 3.2. Problem 2: Data as commodity and effects of personalization as business model (the banality of evil)

Services provided to customers deliver apparent control on their lives and environment and offer a kind of complexity reduction. For instance, the reaction buttons, as the "like" button in several social networks, is an over reduction of emotional reactions.

One of the main trends in the last decades is the concept of personalization, which is driving the data extraction industry in different levels and fields, from commerce to health. This leads to paradoxes like acceptance personalization in services (information), but low acceptance in data collection for personalization [16], which points again to a problem in data governance.

Also, an additional problem is the commoditization of data to generate personalized services, i.e. information just adjusted to personal preferences, leading to a reduction of social cohesion and social atomization, since the preferences of other customers or agents get systematically ignored [17]. The concept of personalization is driving the societies for more isolated consumerism, and thus for more isolated individuals. Personalization is thus discouraging personal contacts and lowers the incentives to create community and discuss with neighbors how to shape the world around as well as to maintain the curiosity for other individuals and cultures; simultaneously personalization promotes the development of echo chambers where individuals reinforce their own perspectives, ignoring alternative points of view.

Data and information are thus contributing to isolate people and create the ideal conditions for totalitarian regimes (who will repress all the liberties – liberty is the possibility to build up communities, belong to these communities and have the possibility to shape things together with other people).

Perhaps this one is the reason why very young people are feeling more and more isolated: In Germany, YouGov surveyed a statistically relevant selection of people in 2019: "How many close friends do you have?" It was defined here as "trusted persons who are close to you". In fact, 11% of respondents ticked "none". And if we assume that a society can only keep moving and function well if the basic needs of individuals are met – then it quickly becomes clear how much the supposedly private issue affects society as a whole. At the same time, it remains a personal one, and one that everyone can help shape. You don't need money or power for that. You just have to look: How can I take care of myself; how can I take care of others? [18].

This development poses a risk of reducing citizens to generators of raw material (data), which is transformed into information for service provision and consumption. In this way, citizens get isolated, only focus on their duties and consumerism. This represents a risk in falling into the banality of the evil [19], a concept coined by Hannah Arend describing the conformism of the German society during the Nazi regime [19]. This represents is a risk for society: there is strong evidence that social networks are not only isolating citizens, but they are also reinforcing within their personalized services a sense of individual helplessness and loss of wellbeing, which

reunite this apparent individual isolation behind dictators and non-democratic and populist political movements [20], [21].

The current trend of personalization is perhaps just helping to generate societies that are vulnerable to manipulation by totalitarian regimes, who are skeptical of traditional communities (like the association with a church or a local community) and who simply foment individuals that commit to their duty and their leisure, without compromise with local communities. A way to come out from this model and problems is by implementing technologies aiming at allowing individual data sovereignty together with forms of social data.

## 4. Solution: Principles for the implementation of social data

Social data as technology (not as a science) should be helpful to reverse this tendency, due the fact that social data can be interlinked from different perspectives. This should provide tools to avoid converting data into raw material. The goal is to define an infrastructure that is exclusively designed for social data and recover the initial intention to generate data to better understand the environment and the world around. This can be implemented in many ways, but all these implementations should allow:

- decentralized storage,
- decentralized data governance, and
- a data structure that allows data-interconnectivity in a straightforward way.

To preserve the social character governance structures are required. This can happen in a micro-scale (single individual - personalization) or a macroscale (large government or enterprise, for instance figuring out how to create a kind of killer app providing all the imaginable services for individuals), but in a meso-scale in order to provide more social cohesion. This because for social data as technology apply a similar principle as for social data as science, i.e. it is possible to understand the human behavior and social interaction through digital traces: through the understanding of human behavior should be possible to understand and decide mechanisms of social cohesion. Thus: *Social Data is used to generate information promoting social cohesion.*

One potential option is the creation of data cooperatives, where small organizations help to define data-governance issues and set up conditions to gather data to provide benefits not only to single individuals but also to local communities [22]. The problem is: in which extend is data and information necessary to shape and help communities to get a better cohesion?

### 4.1. Some hypotheses – data cooperatives

The proposal of data cooperatives as solution to shape communities is based on the following hypotheses:

Hypothesis 1: Data governance in cooperatives occurs in middle-sized organizations with a limited size in order to maintain transparent and equitable conditions to the members. All participants decide on information generation governance politics and goals.

Hypothesis 2: Middle-size organizations avoid individual data governance (which can be challenging for individuals); this can be particularly challenging for elderly patients.

Hypothesis 3: Middle-size organizations should act as non-profit organizations. Middle-size organizations orient themselves to positive shaping society. In some instances, services can be tailored to the individual's business model. In such a case profit-oriented solutions can be derived.

Hypothesis 4: Such organizations can only have a critical size. Fundamental standards, such as data storage and processing, are universal, but goals, solutions, decisions etc. are local. Each smaller organization can be used to generate federated information if this is required.

Hypothesis 5: Middle-sized organizations facilitate open innovation. For given problems solutions are tested on small scales and selected individuals, which represents in some cases a better and safer innovation-incubator than traditional institutional-corporate research.

Hypothesis 6: Social data can also act as a way to implement open and citizen science (democratizing research processes, for instance in health [23]). An additional effect is providing real data, which is completely different from data obtained in experimental conditions [24].

The following are the main challenges of data cooperatives [25]: (1) salary structures; (2) cooperating with other providers and surrounding institutions; (3) building an identity and recruiting potential members; (4) motivation of members to participate actively; and (5) distinction from other types. Benefits are: (1) improvement of conditions; (2) being stronger together; (3) support of research; and (4) data governance. When successful and competent, (data) cooperatives can be powerful tools on public, scientific, and political levels.

## 4.2. FAIR principles as a way to implement social data

The FAIR principle (**Findability, Accessibility, Interoperability, and Reusability** [26]) essentially provides the norms for data in order to make it easy to find (for example with persistent identifiers), easy to access (for example protocols are open and easy to find), make them interoperable (for example using a universal vocabulary and language) and reusable (for example data is shared with a clear and accessible data usage license). Initiatives for data decentralization, like Solid [5] or Gaia-X [6], bring the opportunity to provide an infrastructure compatible to the FAIR principle and capable to make individual produced data into social data. Now, while these initiatives insist on data sovereignty, data privacy and technologies for data governance, such concepts eventually enter in contradiction with the idea of social data, since individual data-sovereignty can stimulate the creation of citizen-centered data-silos, as well as

potential unethical practices of data commerce that could compromise data both protection and ethical parameters for data use.

For this reason, structures and politics are required to promote social data generation from semantic web. Our central hypothesis is that decentralization and data sovereignty alone are not enough to guarantee and provide the parameters (legal, administrative, etc.) promoting the generation and use of social data, and that data-governance in a meta-layer must be promoted, such that data encompasses social needs as well as social natural structures (as shown in part B, Figure 2).

## 5. Potential use cases

One potential solution is the definition of data-cooperatives in health [22]. By gathering social data, it should be possible to improve communities of patients' self-help:

- In oncology: control of patients after treatment, in particular for quality control and to avoid readmission. Also, social data is useful to help patients to adapt their behavior regarding the disease.
- In nephrology: help patients in home-dialysis to correctly perform their dialysis.
- Implementation of patient journeys.

Observe that the implementation of social data requires a different technological basis than those required for the implementation of electronic health records (EHR), which are usually defined and hosted at institutions. In small communities, it is still possible to implement pharmacovigilance solutions. By recording data cooperatives, it is also possible to view the current status of discharged patients. This is often difficult to access due to data protection restrictions: currently in extreme cases hospitals need to analyze advertisements in newspapers about diseased people to record the status and success of performed therapies. In this way, hospitals can also control their quality of provided services. Finally, such solutions should also help patients to be better informed about their health status, reducing in this way the asymmetry between patients and doctors [27]. Finally, data cooperatives are not only focused on one disease with a particular etiology. As data of various natures are collected, from the environment to socioeconomic status, they represent not only an ideal technological basis for disease treatment, but also for prevention, such as early epidemic prevention.

## 6. Conclusion

This short essay analyzes the effects of the structure and methods used in current methods to manage and distribute data. There are in particular two aspects of current data solutions which are particularly problematic: the tendency to design scale-free infrastructures (which oppose natural human relations which are not scale-free) as well as the tendency to provide personalized and customer-centered services. We have provided fundamental definitions for social data and hypothesized how this concept be implemented, for instance on data

cooperatives. Using these principles, we can design strategies for implementing and distributing social data.

## Acknowledgements

We are thankful with Javier Creus for very useful discussions about the concept of data cooperatives, Faizan E Mustafa for comments on a first version of this essay, as well as Stefen Staab for a clarification of the technical background distinguishing data spaces from standard data bases and Martin Grundman for the constant exchange about Solid applied to health care.

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
