# OpenReview forum: "Semantic web, social data as a resource and optimal data governance structures"
_SEMANTiCS.cc/2024/Workshop/NXDG — Submitted to NXDG 2024_

### Official Review · ~Michiel_Fierens1 · 2024-07-26
**Review - Semantic web, social data as a resource and optimal data governance structures**

**Rating:** 5
**Confidence:** 4

**Review:**

The reviewer has determined that the paper does not meet the criteria for acceptance. While the introduction of several interesting hypotheses is commendable and would lead to interesting discussions in the NXDG workshop, the paper would benefit from a more structured approach, a more precise research question, greater consistency and coherence as well as a more in-depth tackling of certain concepts.

Firstly, the text would be more effective if it accurately delineated a particular premise within social data and focused on, say, one specific problem in reality. In this regard, the central hypothesis could, for instance, be based on the premise that the current proposals for decentralisation and data sovereignty are insufficient to drive the generation and utilisation of social data. It would be advantageous to examine in greater detail the underlying structures of companies in conjunction with current trends in personalisation. Alternatively, it would be beneficial to give those underlying structures a more central role in the research to strengthen the argument and provide a contrast with the current proposals for decentralisation and data sovereignty. By encouraging discussion around further research, particularly in the area of semantic web technologies, the researchers can facilitate progress in this field. The introduction of this new structure enables the formulation of more compelling hypotheses and enhances the persuasiveness of the researchers' arguments. Nevertheless, it is not yet clear what the real contribution to the academic debate will be, and the principles and guidelines for future researchers are not yet sufficiently defined.

Secondly, the reviewer has identified instances where potentially interesting arguments or hypotheses for further discussion within NXDG lack logical coherence or are not aligned with the broader structure of the article. Without sufficient coherence, it is challenging to engage the reader in the narrative. In light of the suggested enhanced coherence, it is essential to maintain a clear and consistent link between the research question and the supporting evidence. Several opportunities for improvement are still available here. See for example the link between the semantic web and social data; between social and real data; the consideration of social data as an industrial resource; implications of data being considered a raw material and the origins of these implications being linked to data governance mechanisms not yet explained before; the link between personalisation and isolated individuals; the link between personalisation and totalitarian regimes and social data; the swift introduction of data cooperatives as a solution and how they could improve self-help of patients within communities.

Thirdly, the text does not always provide sufficient clarity on the rationale behind the introduction of specific concepts or terms, nor on their precise meaning and their relationship with the research questions presented. The review indicates that a specific hypothesis or opinion is frequently implied without adequate elaboration or supplementary arguments. In this respect, the text would benefit from additional nuances for the different concepts or terms being introduced such as social data, real data, semantic web developments (Solid or Gaia-X not merely a development in semantic web), contradiction between being a resource and at the same time shaping our lives, restrictive use of data, subjective nature of data, information in small scale being de facto limited, data providers not including public services or governments, the requirement of an initial hypothesis for any form of data production, the swift introduction of mathematical – numerical models and their implications, scale free distribution, log-norm distribution, data as a commodity, meso-scale and lastly the role of data cooperatives and the role of data decentralisation without considering any central elements to enhance e.g. findability of data and its further use like in Data Spaces (Gaia-X example). Implications by the authors now seem to paint quite a black-and-white landscape when this is not always the case.

Finally, the text also contains some grammatical and spelling errors that make it slightly more difficult to read. For example some verbs are not conjugated correctly. See: despite social data is now; individuals gathering and analyse; is a kind of sense of universality; technology apply a similar principle. The text would benefit from a more coherent and fluent sentence structure.

---

### Official Review · ~Rob_Brennan1 · 2024-07-30
**Unfortunately this paper is not worthy of publication in its current form**

**Rating:** 1
**Confidence:** 4

**Review:**

Quality
This paper is poorly structured, written and argued. It contains numerous technical and typographical errors (see below for a sample from the first couple of pages). There is extensive use of low authority sources like webpages and repositories. The definitions provided are used inconsistently. The links to the state of the art seem arbitrary. How does this relate to established studies on data co-operatives?

There is an interesting agenda at the heart of the paper to make data use more accountable and good.

Clarity
There is no coherent research agenda, question or concept laid out in the paper in its current form. The level of English is below that required for publication.
The layout is relatively clean and the figures are well labeled.

Originality and significance
It is hard to rate the originality of the paper given the difficulty in following the argument.
Given the level of basic errors it is not in its current form a significant work.

Detailed Comments

Abstract
Readability suggest
"Despite social data is now an established science to analyze human..."
->
"Despite _the fact that_ social data is now an established science to analyze human..."
typo
"not only require a special"-> "not only require_s_ a special"
typo
"infrastructures of social data" -> "infrastructures _where_ social data"

Introduction
"Data, economics and society are closely related"
Suggest you needed to explain how?

typo "seldom the" -> "seldom _is_ the"
typo "are tacked into account" -> "are ta_ken_ into account"

"Social data is usually considered scientifically, since it promises to learn about “what the world thinks” about a specific issue, brand, celebrity, or other entity."
This is data about people (like personal data). This inconsistent with your defintion in the abstract of social data as "data that is used with a social purpose" which implies to me the "data for good" initiative of the EU.

"When citizens/customers are seen as both resources and people, the concept of data as a resource and the concept of data shaping our lives are contradictory. Semantic-web seem to be a way to solve this contradiction, since i.) it facilitates the use of meta-data (data about data) as semantic annotations, ii.) makes use ontologies to describe knowledge needed to understand collections of web information, and iii.) makes use logic-based techniques to process and query collections of meta-data and ontologies"

This paragraph is extremely incoherent. The properties of "semantic-web" (sic) are not related in any specific way to the proposed contradiction.

"In particular developments in semantic web, like Solid [5] or the Gaia-X [6],"
Gaia-X is not to my knowledge a semantic web initiative.

---

### Decision · Program_Chairs · 2024-08-02

Reject